# Stability of C:N:P Stoichiometry in the Plant–Soil Continuum along Age Classes in Natural *Pinus tabuliformis* Carr. Forests of the Eastern Loess Plateau, China

Haoning Chen [1,†] , Yun Xiang [1,2,†], Zhixia Yao [1], Qiang Zhang [2,*], Hua Li [1,*] and Man Cheng [1]

1 Institute of Loess Plateau/College of Environmental &Resource Science, Shanxi University, Taiyuan 030006, China
2 College of Resources and Environment, Shanxi Agricultural University, Jinzhong 030801, China
* Correspondence: zhangqiang0351@163.com (Q.Z.); lihua@sxu.edu.cn (H.L.); Tel.: +86-13934603466 (H.L.)
† These authors contributed equally to this work.

**Abstract:** Ecological stoichiometry is useful for revealing the biogeochemical characteristics of flows of nutrients and energy between plant and soil, as well as the important implications behind these ecological cycling phenomena. However, the ecological stoichiometric linkages among leaf, litter, soil, and enzymes in the natural forests of the Loess Plateau remain largely unknown. Here, leaf, litter, and soil samples were collected from four age classes of natural *Pinus tabuliformis* Carr. (*P. tabuliformis*) to explore the deep linkages among these components. We measured the total carbon (C), total nitrogen (N), and total phosphorus (P) concentrations of leaf and litter, as well as the concentrations of soil organic C, total N, total P, nitrate N, ammonium N, available P, and the activities of β-1,4-glucosidase (a C-acquiring enzyme), β-1,4-N-acetylglucosidase (an N-acquiring enzyme), and alkaline phosphatase (a P-acquiring enzyme) in the topsoil (0–20 cm). The average leaf N:P was 6.9 indicated the growth of *P. tabuliformis* was constrained by N according to the relative resorption theory of nutrient limitation. The C:N, C:P, and N:P ratios in leaf, litter, and soil and the enzyme activity were not significantly different among age classes ($p > 0.05$). Litter C:N (43.3) was closer to the ratio of leaf C:N (48.8), whereas the litter C:P (257.7) was obviously lower than the ratio of leaf C:P (338.15). We calculated the stoichiometric homeostasis index (1/H) of leaf responses to soil elements and enzyme activities and found that the relationship between leaf C:P and soil C:P was homeostatic ($p < 0.05$), whereas the remaining indices showed the leaf stoichiometries were strictly homeostatic ($p > 0.05$). Correlation analysis showed both litter C:P and N:P were positively correlated with leaf and soil C:P, while the stoichiometric ratios of soil elements and enzymes were obviously irrelevant with leaf stoichiometries ($p > 0.05$). Partial least squares path modeling indicated that litter significantly changed soil element and enzyme characteristics through direct and indirect effects, respectively. However, soil elements and enzymes impacted leaf stoichiometries barely, which was further confirmed by an overall redundancy analysis. In summary, C:N:P stoichiometry within the plant–soil continuum revealed that natural *P. tabuliformis* is a relatively stable ecosystem in the Loess Plateau, where the element exchanges between plant and soil maintain dynamic balance with forest development. Further studies are needed to capture the critical factors that regulate leaf stoichiometry in the soil system.

**Keywords:** C:N:P stoichiometry; natural *Pinus tabuliformis* Carr.; plant–soil continuum; age classes

## 1. Introduction

Ecological stoichiometry is an effective method for understanding the balance and interactions of energy and multiple chemical elements under different levels of biogeochemical cycling [1,2]. All living things are composed of specific proportions of elements [2] and tend to maintain dynamic balance through feedback links with their environment. Thus, organism stoichiometries are widely applied as critical indicators of ecological processes

and functional stability [3–5]. Both nitrogen (N) and phosphorus (P) are major limiting elements in governing plant growth and various physiological activities in terrestrial ecosystems. The leaf N:P ratio can be used to diagnose the soil nutrient supply status in diverse multiscale systems [6,7], but the threshold ratios for N or P limitation remain inconclusive depending on plant species, stages, and organs [3]. With further exploration in the field of stoichiometry, the bulk of studies have focused on the characteristics of plants corresponding to the inconsistent environment [8], in which community stability, functional diversity, and productivity were encompassed [1,9]. These studies provided new perspectives for ecological stoichiometry.

The balance of C, N, and P within ecological interactions is the focus of ecological stoichiometry [2]. The properties and functions of ecosystems can be grandly affected by feedback mechanisms between above- and belowground components [10,11], which are closely associated with spatial and nutrient structure in ecosystems. Previous studies [12–14] have demonstrated that soil resource pattern and stoichiometric ratios may vary considerably among plant community restoration, development types, and successional processes. Plants release substrates to soil primarily in the form of litter and root exudates, and litter breakdown in soil can improve soil fertility and mediate microbial nutrient limitation [15,16]. Some studies have shown that litter serves as a nexus for linking plant and soil [17,18]. However, the above studies only focused on nutrient flows from leaf to soil; nevertheless, the processes of plants exporting substrates to and absorbing nutrients from soil should be mutually coupled [19] (Figure 1). The plant-derived organic matter trapped in the soil is converted into bioavailable small molecule nutrients by microbial intervention, subsequently ensuring continued plant production and ecological functions. Many physiological activities of plants are regulated by soil properties, such as the difference in resorption rates for various elements in nutrient-poor soil [20]. Fan et al. [21] and Chen et al. [22] reported that the plantations gradually converted to a P limitation along a chronosequence because soil N:P increased and available P content decreased. Xiao et al. [23] conducted a secondary succession research in abandoned grasslands and revealed that eco-enzymatic activities and stoichiometric ratios played significant roles in manipulating the plant tissue stoichiometries and species replacement. Consequently, a better knowledge of elemental coupling in the plant–soil continuum using ecological stoichiometry will assist in improving the prediction of interactions between multiple elements in diverse ecosystems [24].

The Loess Plateau in China, one of the most severely degraded and sensitive ecosystems in the world, is subject to various complex ecological problems [25]. Benefiting from the implementation of the "Grain-for-Green" project in 1999, most of the steep-slope croplands were returned to woodland and grassland in this region [26,27], providing rich platforms for exploring the diversely ecological interactions between plant and soil. Recently, ecological stoichiometry studies within the plant–soil continuum under artificial management have become hotspots at regional scales [11,22]. These investigations can facilitate the reasonable use of forest resources and predict climate change as a result of changes in the soil C pool. However, Bai et al. [28] compared the difference in stoichiometric homeostasis of plant organs among four kinds of forests and found that natural *Quercus wutaishansea* was superior in adapting to water-deficient and nutrient-poor soil. Indeed, native forests are more resilient than younger plantations in terms of C storage, soil erosion control, and biodiversity benefits in hostile environments [29]. Therefore, studies with attention on changes in ecological stoichiometry with natural forest development will fill the gap in the natural ecosystem research and contribute to further understanding of underlying ecological stability mechanisms on the Loess Plateau.

Four age classes of natural *P. tabuliformis* forest were chosen for our study on the Loess Plateau. Plant leaf, litter, soil element, and extracellular enzyme were determined to explore their relations within C, N, and P stoichiometry. We hypothesized the following: (i) the C:N:P ratios of plant leaf, soil element, and enzyme activity vary with age classes; (ii) litter C:N or C:P would be greater than that of leaves throughout the age sequence because of

the nutrient conservation mechanism of plants [6]; (iii) the element stoichiometries would be tightly correlated between soil and leaf, and continuous processes of element exchange would occur between them via litter and enzymes.

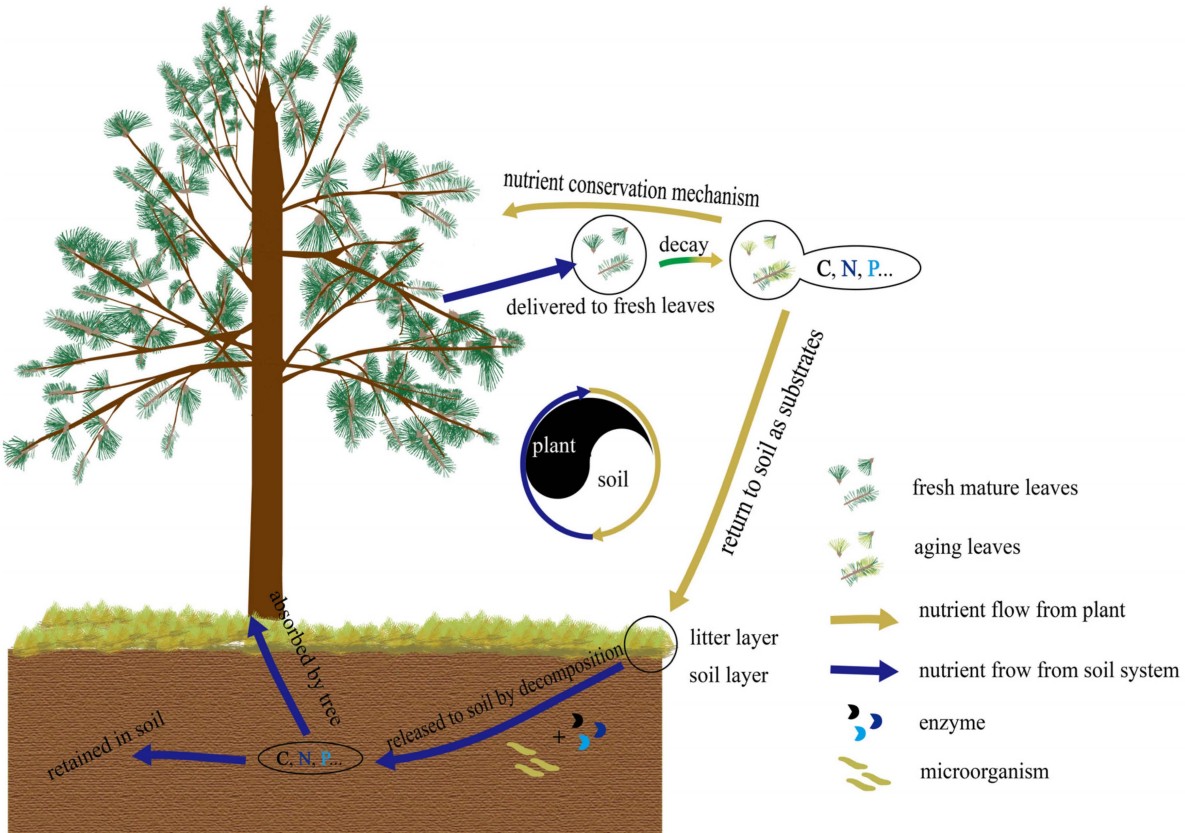

**Figure 1.** Graphs of element exchange between plant and soil in our study.

## 2. Materials and Methods

### 2.1. Study Sites

The research was carried out in Shanxi Province, eastern Loess Plateau (110°14′–114°33′ E, 34°34′–40°44′). The region has a total area of 156,806 km$^2$ and belongs to a temperate and semi-arid continental climate. The average annual temperature and average annual precipitation are 4–14 °C and 400–600 mm, respectively, with a frost-free period of 120 to 220 d. Heavy rainfall concentrated from July to September accounts for 70% of the annual precipitation. More than 80% of the area is occupied by mountains and hills, with an elevation of 1200–2000 m. The forest coverage rate is approximately 20%, and the main soil type is classified as Alfisols developed from Malan loess [30].

### 2.2. Experimental Design

We selected 23 experimental sites from the field in October 2020 (Figure 2). Our research focused on the ecological stoichiometric characteristics of the C, N, and P balance in *P. tabuliformis* stands of different ages. Based on the research of stand ages being strongly connected to tree diameter at breast height (DBH) as showed by Dey [31], we classified *P. tabuliformis* forests into four continuous diameter classes according to their DBH and explained age structures by diameter class [32]. The lowest DBH was 11.75 cm in this research and was split into the following diameter classes: 10 < DBH ≤ 15 cm, 15 < DBH ≤ 20 cm, 20 < DBH ≤ 25 cm, and 25 < DBH ≤ 30 cm, corresponding to Age Classes (ACs) I, II, III, and IV. Basic information on the four ACs is shown in Table 1.

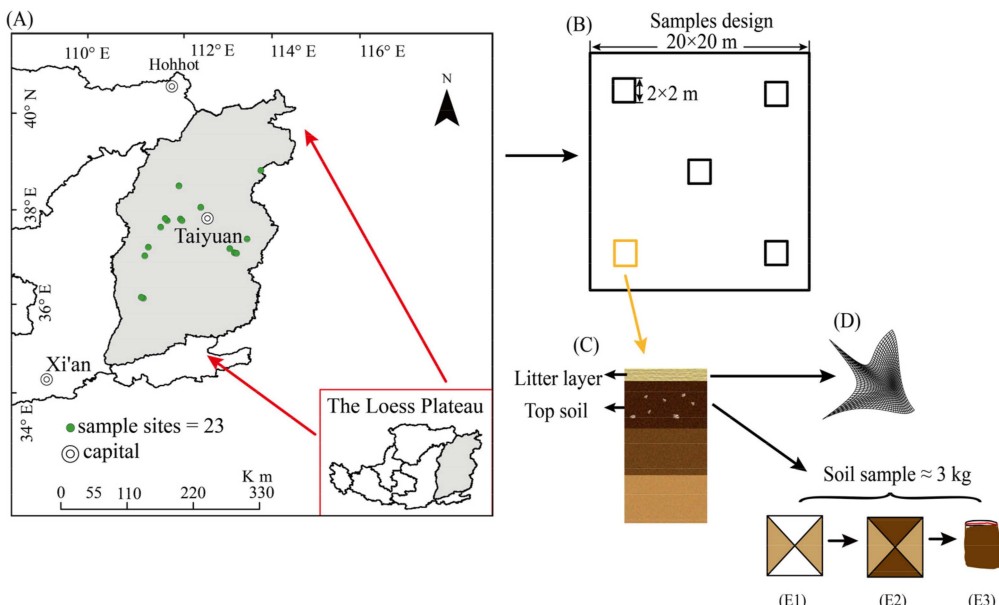

**Figure 2.** Graphs of sample collection for this research. Due to the limited area of natural *P. tabulae-formis* forests in Shanxi Province, certain sample sites were near together or overlapped in the plot. (**A**) Study area and distribution of sample sites; (**B**) five subquadrats were set up within each site; (**C**) litter samples were collected from the litter layer and loaded into the tuck net; (**D**) soil samples were collected from the top soil layer (0–20 cm), mixed and removed from the lighter colored areas according to the diagonal method repeated twice (E1, E2) to form the final soil sample (E3).

**Table 1.** Basic information regarding different ACs of samples.

| ACs | Mean Elevation (m) | Mean Slope Gradient (°) | Mean Stand Density (Tree·ha$^{-1}$) | Mean DBH (cm) | Mean Height (m) |
|---|---|---|---|---|---|
| I (4) | 1359.05 ± 218.89 | 46.25 ± 14.36 | 138 | 13.81 ± 1.38 | 10.55 ± 0.40 |
| II (12) | 1365.22 ± 141.69 | 46.17 ± 12.55 | 208 | 17.09 ± 1.34 | 10.76 ± 1.75 |
| III (5) | 1363.15 ± 285.98 | 52.80 ± 11.80 | 110 | 24.35 ± 0.60 | 9.81 ± 3.52 |
| IV (2) | 1541.56 ± 68.24 | 35.50 ± 7.78 | 150 | 29.75 ± 0.35 | 13.50 ± 0.71 |

The number of sample sites (n) are displayed along with each AC in the first column.

### 2.3. Sampling and Determination

A 20 × 20 m sampling plot was set up in the middle slope of each *P. tabuliformis* forest site. Within each sampling plot, approximately 500 g of mixed mature and healthy leaves was collected from the east, west, south, and north-facing sides at a distance of 4–5 m from the ground using high branch scissors. Five subquadrats (2 × 2 m) were set up in the four corners and center of each sample plot. The litter was collected and mixed into one breathable tuck net. Afterward, we removed the remaining litter and collected topsoil (0–20 cm) from five quadrats with a soil auger (diameter = 50 mm) and combined them into one soil sample (3 kg). The same approach was carried out on each sampling plot.

The leaf and litter samples were dried to constant weight at 70 °C. The soil samples were divided into two parts after removing the visible plant residues and stones. One part was air-dried and screened with a 0.149 mm sieve for basic chemical property analyses. The other part was stored in the refrigerator (4 °C) for extracellular enzyme activity testing.

The acid-dichromate FeSO$_4$ titration method modified by the Walkley–Black method was used to measure plant total C and soil organic C (SOC) concentrations [33]. After digestion with H$_2$SO$_4$ and H$_2$O$_2$, the plant total N and soil total N (STN) concentrations were determined using the Kjeldahl method. The plant total P and soil total P (STP) concentrations were measured using the molybdenum antimony reagent colorimetric method. The soil available P (SAP) was measured using the Olsen method [33]. The soil

ammonium N (SAN) and nitrate N (SNN) were measured using a continuous flow analyzer (Auto Analyzer 3, SEAL, Germany).

The activities of soil C-acquiring (β-1,4-glucosidase, BG), N-acquiring (β-1,4-N-acetyl-glucosidase, NAG), and P-acquiring (Alkaline phosphatase, ALP) enzymes were determined using a simplified microplate fluorometric assay described by Saiya-Cork [34]. Adding 1 g of soil to 125 mL 50 mM sterile sodium acetate buffer and homogenizing 10 min at 180 r min$^{-1}$ to prepare soil suspensions. Suspensions of 200 μL were mixed, respectively, with 50 μL 200 μM specific fluorometric substrate of each enzyme and 50 μL buffer into 96-well microplates to form three analytical replicates and one control for each sample. Afterwards, the microplates of ALP enzyme and the other two enzymes were incubated at 25 °C for 2 h and 4 h under dark condition, respectively. Finally, the fluorescence was measured using a microplate fluorometer at 365 nm excitation and 450 nm emission filters. The eco-enzymatic activity was calculated using the DeForest method [35]. The enzyme stoichiometric ratios of BG:NAG, BG:ALP, and NAG:ALP were calculated using ln(BG):ln(NAG), ln(BG):ln(ALP), and ln(NAG):ln(ALP), respectively. The detailed calculations of vector angle and vector length referred to the Moorhead et al. method [36] to indicate the intensity of microbial nutrient and C limitation, respectively.

### 2.4. Data Analysis

Statistical analyses were performed using Excel 2019, SPSS 19.0 (SPSS, Inc., Chicago, IL, USA) and R v.4.1.3 (https://cran.microsoft.com/snapshot/2022-03-25/bin/windows/base/). We used one-way analysis of variance to assess the variations among different ACs. The least significance difference (LSD) test was organized at 95% confidence level ($p < 0.05$) to examine the significant differences of sample means suited normal distribution. Additionally, for non-normal data, we conducted the Kruskal–Wallis rank test to analyze the differences. The Spearman coefficients were calculated to test the significant correlations between plant leaf, litter, soil element, and enzyme activity at a false discovery rate (FDR) corrected value of $p < 0.05$. The "corrplot" package in R was used to form the correlation figure. To explore the further relationships between plant and soil, the path coefficients were calculated through partial least squares path modeling (PLS-PM) implemented by the "plspm" package in R. A redundancy analysis (RDA) was conducted using the "vegan" package to calculate the impact of soil elements, enzyme activities, and their stoichiometries on leaf stoichiometries, based on Monte Carlo permutations (permutation = 999).

Stoichiometric homeostasis, the essential concept in ecological stoichiometry, can well reflect organisms' physiological and biochemical adaptability to inconstant resource environments [1,8,19]. Consumers might establish tight associations with resources through complex and integrated ecological activities. Similarly, through a variety of physiological mechanisms, consumers can maintain the relative stability of their own elemental composition and environmental element cycling [24]. As a result, the homeostasis theory can be properly applied to ecosystems in which consumers (such as plants) and resources (such as soil substrates) interact closely, especially in forests [28,37]. To assess the degree of homeostasis of natural *P. tabuliformis* forests developed along ACs, we calculated the homeostatic index (1/H). As reported by Makino et al. [38], the 1/H value was derived from the following equations:

$$Y = c \times X^{\frac{1}{H}} \tag{1}$$

where c is a fitted constant, Y represents the leaf stoichiometry, and X represents the soil element and enzyme activity stoichiometry in our research. For easy understanding, Equation (1) can be log-transformed to linear form:

$$\log(Y) = \log(c) + \frac{1}{H}\log(X) \tag{2}$$

In Equation (2), 1/H will trend to zero infinitely when $\log(Y) \approx \log(c)$, then the homeostatic relationship is defined as strict homeostasis [38]. Furthermore, the regres-

sion relations between Y and X will gradually fade to insignificant level ($p > 0.05$). This postulation is also acceptable to Equation (1). However, if the regression relationship is significant, the degree of homeostasis can be classified as: $0 < 1/H < 0.25$ (homeostatic), $0.25 < 1/H < 0.5$ (weakly homeostatic), $0.5 < 1/H < 0.75$ (weakly plastic), and $0.75 < 1/H$ (plastic) [39]. Here, we fitted Equation (1) to calculate $1/H$ values.

## 3. Results

### 3.1. Ecological Stoichiometry in the Plant–Soil Continuum

As shown in Figure 3, the SOC, STN, and STP concentrations generally increased with AC. The SAN and SNN concentrations in AC IV decreased by 94.9% and 50.4% compared with AC I, respectively. Analysis of variance exclusively indicated significantly higher SAP concentrations at the AC IV stage compared with other ACs ($p < 0.05$). Compared with AC I, both the enzyme activities of BG and NAG increased by 43.0% at a non-significant level under AC IV stage, and ALP activity was similar to that of AC I ($p > 0.05$).

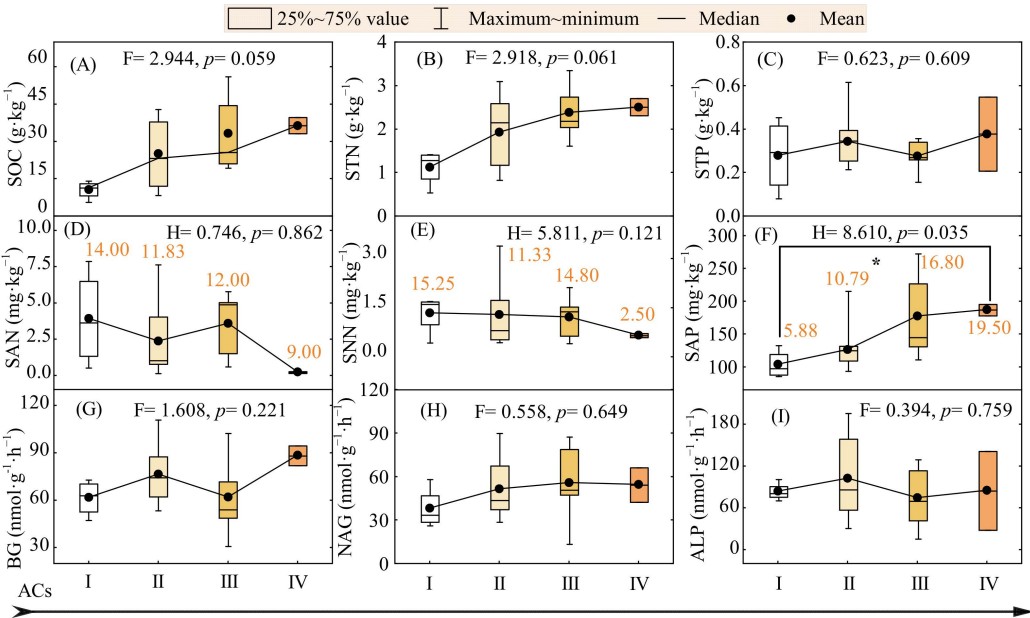

**Figure 3.** Characteristics of soil nutrient concentrations (**A–F**) and extracellular enzymatic activity (**G–I**) changed with ACs in the surface soil. (*) indicate the significant difference between different ACs. The brown-red letters represent mean ranks of each AC in subfigures (**D–F**), which were calculated using the Kruskal–Wallis rank test. The differences among four ACs in other subfigures were examined by the LSD test.

The highest leaf C:N ratio was observed in AC I (52.72, Figure 4A). Litter C:P and N:P reached the maximum values (329.0 and 6.4, Figure 4E,F) at the AC IV stage. Likewise, soil C:N, C:P, and N:P values were highest in AC IV as shown in Figure 4G–I. Overall, plant leaf, litter, and soil C:N, C:P, and N:P ratios were similar with ACs and exhibited no significant variation ($p > 0.05$). In terms of enzyme stoichiometries (Figure 4J–L), compared with AC I, BG:ALP increased from 0.93 to 1.11, and NAG:ALP increased from 0.81 to 1.00 in the AC IV stage. In general, the enzyme stoichiometry BG:NAG, BG:ALP, and NAG:ALP ratios demonstrated no difference across ACs ($p > 0.05$).

The vector angle and vector length were calculated to indicate the intensity of microbial resource limitation (Figure 5A,B). Greater deviations of vector angle from 45° indicates stronger N or P limitation, and longer vector length indicates severer microbial C limitation. Compared with AC I, vector angle decreased by 13.5%, while vector length increased by 13.6% at the AC IV stage. These results showed that P limitation of microbial metabolism was gradually alleviated and C limitation slightly increased as stand age increased.

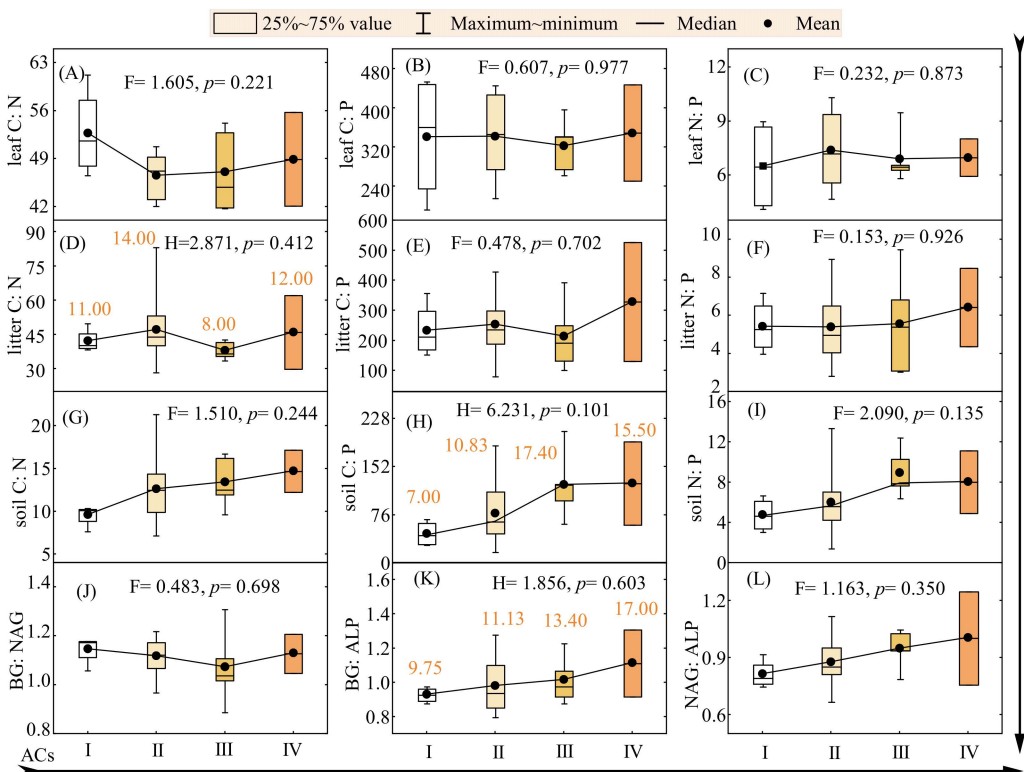

**Figure 4.** Ecological stoichiometric ratios (mass ratios) for plant leaf (**A–C**), litter (**D–F**), soil nutrient (**G–I**), and extracellular enzymatic activity (**J–L**) along ACs.

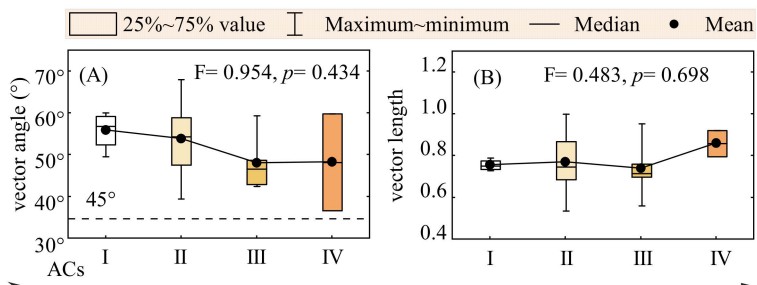

**Figure 5.** Eco-enzyme stoichiometry indicated nutrient limitation (vector angle, **A**) and carbon limitation (vector length, **B**) of microbial metabolism.

### *3.2. Stoichiometric Relationships within the Plant–Soil Continuum*

The correlations among ecological stoichiometry are shown in Figure 6A. Both leaf and soil C:P were positively correlated with litter C:P and N:P. The activity of NAG was positively correlated with SOC and STN. Both soil C:P and N:P showed significantly positive correlations with the enzyme NAG:ALP and negative correlations with the vector angle. The stoichiometric ratios of soil element and enzyme were irrelevant with leaf stoichiometries. SAN was negatively correlated with leaf C:P and leaf N:P.

In addition, we used PLS-PM to explore the pathway coefficients among leaf, litter, soil element, and enzyme characteristics (Figure 6B). The blue arrows showed that litter had significantly direct impact on soil element characteristics and indirect impact on enzyme characteristics. The orange arrows indicated that soil element and enzyme characteristics did not significantly affect leaf stoichiometries.

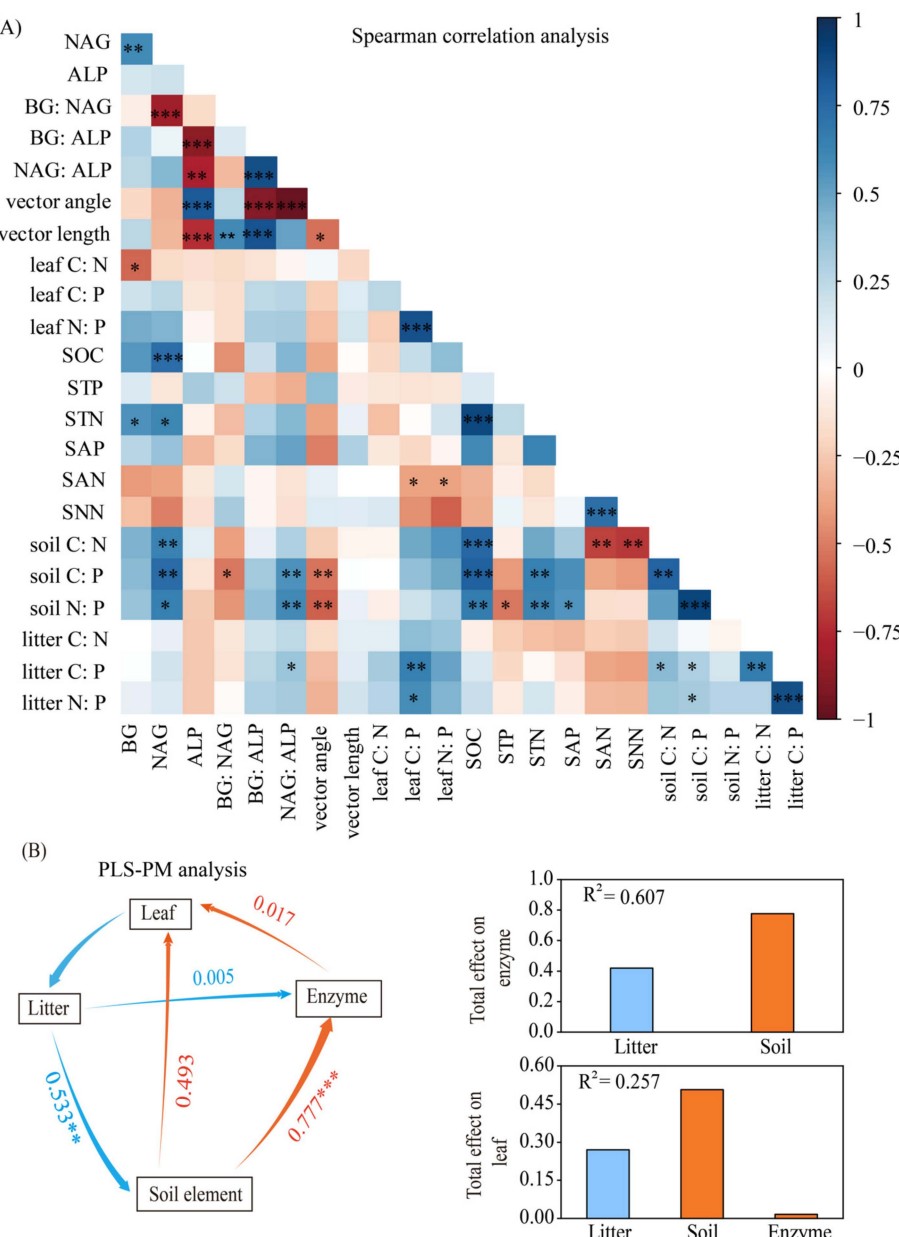

**Figure 6.** Relations of stoichiometry within the plant–soil continuum. (**A**) Correlations of stoichiometry among plant leaf, litter, soil, and enzyme activity. (**B**) Pathway effects between plant and soil system (soil element and enzyme) calculated by partial least squares path modeling (PLS-PM). The blue arrows represent the effect of litter on soil element and enzyme activity stoichiometries, and the orange arrows indicate the effect of soil system on leaf stoichiometries. (*, **, and ***) represent significant correlations at 0.05, 0.01, and 0.001 levels, respectively.

Since the potential effect of litter was included in the path coefficients along the soil toward the leaf, we used the RDA to investigate the exclusive impact of soil on the leaf while removing this interference. The results showed that constrained variables explained 48.73% of total variations in leaf stoichiometries, with Axis 1 and Axis 2 accounting for 33.45% and 15.22%, respectively (Figure 7). A Monte Carlo test with 999 permutations showed that SAN and soil BG activity had significant effect on leaf stoichiometries ($p < 0.01$), but the overall RDA result was not significant ($p = 0.359$).

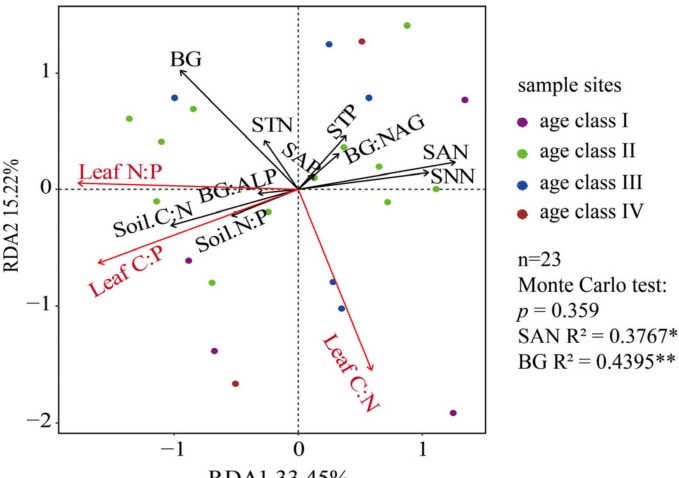

**Figure 7.** Redundancy analysis (RDA) showing the relationship between soil system variables and leaf stoichiometries. The overall RDA result was not significant with 999 permutations (*p* = 0.359). The constrained variables explained 48.73% of the total variance. (**) represent significant correlations at 0.01 level.

Furthermore, we calculated the stoichiometric homeostasis indices for the leaf to explore the homeostatic degree of the plant (Table 2). The 1/H indices between the leaf and the stoichiometric ratios of soil element and enzyme were "SH" (*p* > 0.05), suggesting a strictly homeostatic relationship (SH), except for the relationship between soil C:P and leaf C:P (H: homeostatic, *p* = 0.031, $R^2$ = 0.157).

**Table 2.** Stoichiometric homeostatic relationships between plant leaf and corresponding soil and enzyme stoichiometry.

| Variable | | Fitting Equation | Homeostasis Indices (1/H) | $R^2$ | *p* |
|---|---|---|---|---|---|
| **X** | **Y** | | | | |
| soil C: N | leaf C: N | $Y = 54.134 \times X^{-0.049}$ | −0.049 SH | 0.013 | 0.291 |
| soil C: P | leaf C: P | $Y = 174.026 \times X^{0.154}$ | **0.154 H** | 0.157 | 0.031 |
| soil N: P | leaf N: P | $Y = 5.554 \times X^{0.136}$ | 0.136 SH | 0.064 | 0.131 |
| BG: NAG | leaf C: N | $Y = 49.127 \times X^{-0.061}$ | −0.061 SH | 0.033 | 0.183 |
| BG: ALP | leaf C: P | $Y = 340.216 \times X^{0.068}$ | 0.068 SH | 0.028 | 0.231 |
| NAG: ALP | leaf N: P | $Y = 7.438 \times X^{0.092}$ | 0.092 SH | 0.052 | 0.158 |

Bold indicates significant relationship (*p* < 0.05). SH represents strictly homeostatic; H represents homeostatic.

## 4. Discussion

### 4.1. Ecological Stoichiometry of C, N, and P in Plant Leaf and Litter

N and P are the most essential elements in constructing plant tissues and maintaining growth [40]. The C:N:P ratio is often connected with multiple functional processes such as nutrient cycling and reutilization among the components of ecological systems [1,41], and thus, can be used as a powerful tool to investigate the interactions of elements in the plant–soil continuum [1,42]. In the current research, the ratio of plant leaf C:N (average value = 48.8) was higher than the ratio calculated for the global leaf average C:N (37.37) [6]. Zhang et al. [43] reported the adaptive growth hypothesis of plant survival strategy along environmental gradients during the evolution process. This viewpoint emphasized that higher plant C:N enables survival through higher N-use efficiency (survival priority strategy) under severely N-limited conditions, while a lower C:N ratio means that the plant sustains rapid growth through favorable competition (growth priority strategy) under a suitable environment. This means that organisms with larger C:N ratios have a higher N-use efficiency [44]. As shown in Figure 4C, all ACs of *P. tabuliformis* were observed under severe N limitation (refer to para. 2 of 4.1). Consequently, the relatively higher leaf C:N

ratio in this region might indicate that more efficient N utilization to ensure plant durable growth in current soil conditions.

The average leaf C:P and N:P were 338.15 and 6.95, respectively, which were smaller than the worldwide average leaf ratios (C:P = 516.39, N:P = 12.57) [6]. These results were consistent with previous findings [28], suggesting that *P. tabuliformis* leaves in this region had a high soil P absorption rate. Plant growth depends on the synthesis of proteins by ribosomes, which requires large amounts of N and P supply [45]. Aerts and Chapin [46] suggested the adaptation of plants to N and P nutrient deficiencies was reflected in shifts of the leaf N:P ratio. Consequently, we can utilize N and P concentrations and N:P ratio as definite indicators to assess nutrient limitation [1,47,48]. However, there are no unanimous thresholds for identifying nutrient limitation types because of interacting factors such as environmental conditions and species characteristics [3,47]. Recently, emerging research has attempted to diagnose the type of nutrient limitation through leaf N and P resorption efficiency [20,49]. According to the relative resorption hypothesis proposed by Han [20], the boundary N:P ratio in conifers was 8.96, below which plants face severe N limitation and develop a higher N resorption. Based on the above previous research, we inferred that the growth of natural *P. tabuliformis* was hampered by N because of the lower leaf N:P (6.95). This result was supported by earlier studies on the Loess Plateau [22,50].

After being distributed among plant tissues, bioavailable nutrients are returned to the soil using litter as a carrier [51]. Litter, a key soil fertility supplement, serves as a continuous bridge between plant and soil [15,52]. Therefore, investigations on litter stoichiometric ratios could help clarify the characteristics of material cycling within the plant–soil continuum [1,6,15]. In the present research, the average litter C:N (43.3) was close to leaf C:N (48.8), whereas litter C:P (257.7) was obviously lower than leaf C:P (338.15). These results did not support Hypothesis (ii) and differed considerably from previous findings [6,17]. Furthermore, as shown in Figure 8, we calculated the total C content in litter was 44.36%, which was significantly lower than the leaf total C content (55.31%). The total N content in the litter was 1.03%, which was close to the incompletely absorbed N value (1.00%), while the total P content in the litter was 0.21%, considerably higher than the incompletely absorbed *p* value (0.08%) [53]. These results supported Han's theory [20] and revealed that the similar litter C:N and lower C:P compared to leaf ratios might relate to faster C loss, higher P retention in litter, and a relatively higher N resorption rate than P of leaf under N-poor soil conditions. Comparing with the global litter average C:N (56.74) and C:P (1218.25) [6], the lower C:N and C:P in this region might suggest that litter decomposes rapidly and, thus, nutrients are exchanged more frequently in the plant–soil continuum, compensating for the suppressive effect of plant growth caused by N limitation. These findings indicated that *P. tabuliformis* forests have evolved multiple positively modified mechanisms to adapt to long-term N-limited conditions [24].

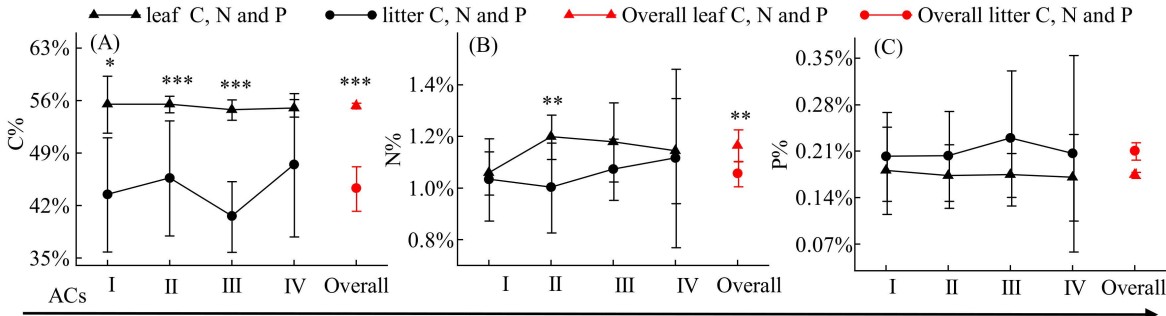

**Figure 8.** The C (**A**), N (**B**), and P (**C**) content in leaf and litter. The black symbols represent the C, N, and P content at each AC, and the red symbols represent the overall content in leaf and litter. (*, **, and ***) signify content differences between leaf and litter that are significant at 0.05, 0.01, and 0.001 levels, respectively.

*4.2. Ecological Stoichiometric Characteristics among ACs*

The previous literature shows that, along with increasing stand ages, the allosteric accumulation or loss of C, N, and P concentrations in tree leaf, litter, and soil induced the changes in the C:N:P ratio [54,55]. Contrary to Hypothesis (i), we found no significant variation in leaf, litter, and soil C:N, C:P, and N:P ratios across ACs in the natural *P. tabuliformis* forests, suggesting this ecosystem is relatively stable [22], which is beneficial for the dynamic balance of element exchange between the plant and soil system [24]. The homeostatic degree of dominant species is highly connected to ecosystem structure, function, and biomass stability [1,9]. Therefore, we calculated the stoichiometric homeostasis indices of leaves (Table 2), and most results were categorized as "SH", implying that the plant may change the availability and utilization efficiency of the limiting element modified by a variety of physiological pathways to stabilize the stoichiometric ratios of the organism [24]. Even under long-term N-limited condition, such intrinsic self-regulating mechanisms assisted in maintaining the relative stability of productivity and elemental turnover in forests. However, a recent study by Wang et al. [37] showed that stoichiometric homeostasis was coupled with tree growth and mature trees sustained stronger leaf N:P homeostasis than young trees. As shown in Table 1, the DBH of the studied trees was at least 10 cm, which belonged to the transition phase from middle-age to over-mature forest [56] and might imply that the sensitivity of plant response to nutrient dynamics turns conservative gradually [8]. Therefore, a major limitation in this study is being unable to investigate the characteristics of stoichiometric variation in natural *P. tabuliformis* forests through a complete age sequence.

Likewise, variations in soil enzyme stoichiometry were not significant among ACs, which might be attributed to the steady environment in this region. An average enzymatic vector angle > 45° showed that the microbial metabolism encountered P limitation [35]. As the forest progressed to a higher age class stage, the enzymatic vector angle decreased and approached to 45° gradually, indicating that microbial P limitation was alleviated (Figure 5A). Extracellular enzyme synthesis largely depends on environmental simulation [57], both nutrient deficiency and substrate abundance may drive enzymatic activity [57,58]. As mentioned above, a large proportion of P was retained in litter as incomplete absorption. Litter released P into soil following decomposition, which facilitated the replenishment of SAP toward high ACs (Figure 3F, $p < 0.05$) and alleviated microbial P limitation [16]. This suggested that there were complex and intimate linkages among the components across the plant–soil continuum. The nutrient flow from plant to soil facilitated the benign maintenance of microbial survival in the soil system.

*4.3. Relationships among Ecological Stoichiometric Ratios in the Plant–Soil Continuum*

During forest development, frequent interactions between plant and soil components may strongly impact the stability of ecosystem functions [59]. Litter derives from leaf, a major natural source of soil nutrient compensation [15,18], and extracellular enzymes in the soil can hydrolyze complex organic matter into tiny molecules with high biological accessibility [60]. The coupling of these two processes together modulates the material cycle of the ecosystem. Therefore, it is necessary to explore the deeper linkage of C, N, and P stoichiometry among different parts within the plant–soil continuum [1,61].

In this study, we found that both litter C:P and N:P were positively correlated with leaf and soil C:P, and litter C:P was positively correlated with soil C:N (Figure 6A). Moreover, the effect of litter on enzymatic stoichiometry was primarily related to soil characteristics (Figure 6B). These results supported the findings of previous studies [17,18] and partially supported Hypothesis (iii) that litter acts as a medium for nutrient transmission from the leaf to soil pool.

Some previous studies based on rhizosphere soils have shown that enzyme activity and stoichiometry were essential factors determining mineral elements in plant tissues [23,62]. However, correlation analysis suggested that soil element and enzyme stoichiometric ratios were not significantly associated with leaf stoichiometry (Figure 6A). Furthermore, the

pathway and overall RDA analysis demonstrated that elemental flow derived from soil and enzyme directions had no significant effect on the leaf (Figures 6B and 8). These findings did not support Hypothesis (iii) and may be caused by multiple factors. The leaf stoichiometric characteristics are more likely to be a combination of nutrient transport within the plant organism and strictly inherent self-regulating mechanisms (Table 2) [24,51]. In addition, other studies have revealed that microbial biomass, community composition, and diversity exert significant impact on the mineral nutrients present within plants [63,64]. In this study, inadequate selection of parameters or analysis model settings may have decreased the evaluated accuracy of the relationship between the soil system and leaf stoichiometry. Therefore, further work is needed to elucidate the contribution of microbial characteristics to the shifts of elemental stoichiometry within the plant–soil continuum on the Loess Plateau.

### 5. Conclusions

In this study, we explored the C, N, and P stoichiometric characteristics within the plant–soil continuum with ACs on the Loess Plateau. Based on the relative reabsorption theory of leaf N:P, the growth of *P. tabuliformis* was limited by N (leaf N:P = 6.9). The average leaf C:N was similar to litter C:N, whereas leaf C:P was substantially higher than litter C:P because higher P was retained in the litter, indicating a relatively higher N resorption under N-limited conditions. Then, the stoichiometric ratios of leaf, litter, soil element, and enzyme activity did not vary with AC, and most of the leaf stoichiometric homeostasis indices were categorized as "SH". These results revealed that *P. tabuliformis* forests are relatively stable, and the trees utilize nutrients with a more conservative and self-regulatory strategy. In addition, both litter C:P and N:P were positively correlated with leaf and soil C:P, and litter stoichiometric ratios indirectly modulated enzyme activity. The above results suggested litter may work as a nutrient hub between leaf and soil systems. In contrast, the characteristics of soil element and enzyme activity exerted little impact on leaf stoichiometry. These findings provide an improved understanding of nutrient cycling and ecological stability mechanisms in the natural forest on the Loess Plateau.

**Author Contributions:** Conceptualization, H.C., Y.X. and M.C.; methodology, H.C., Z.Y., Y.X. and M.C.; formal analysis, H.C. and Z.Y.; investigation, H.C., Z.Y., Y.X. and M.C.; resources, Y.X., M.C., H.L. and Q.Z.; data curation, Y.X. and M.C.; writing—original draft preparation, H.C., Y.X., H.L. and Q.Z.; writing—review and editing, H.C., Y.X., H.L. and Q.Z.; visualization, H.C. and Z.Y.; supervision, Y.X., M.C., H.L. and Q.Z. All authors have read and agreed to the published version of the manuscript.

**Funding:** This research was funded by the Natural Science Foundation of Shanxi Province (202203021 211317), the project of scientific and technological innovation of Shanxi Agricultural University (2020BQ16), and the postdoctoral program of Shanxi province.

**Institutional Review Board Statement:** Not applicable.

**Informed Consent Statement:** Not applicable.

**Data Availability Statement:** The data that support the findings of this study are available on request from the author (H.C.). The data are not publicly available due to privacy or ethical restrictions.

**Conflicts of Interest:** The authors declare no conflict of interest.

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
