# Peer review of "Stability of C:N:P Stoichiometry in the Plant–Soil Continuum along Age Classes in Natural Pinus tabuliformis Carr. Forests of the Eastern Loess Plateau, China"

_forests, doi:10.3390/f14010044_

Round 1

Reviewer 1 Report

Review:

Stability of C:N:P stoichiometry in the plant–soil continuum along age classes in natural Pinus tabuliformis forests of the eastern Loess Plateau, China

Chen et al., 2022, Forests

This manuscript reports on the ecological stoichiometry (C:N:P) of leaf, litter and soil within 23 red pine stands, classified into four different diameter classes, using diameter at breast height as a proxy for age, to determine the degree of homeostasis within Pinus tabuliformis. The paper draws several conclusions, showing that leaf growth is largely strictly homeostatic according to values of homeostatic indices, that litter and soil stoichiometry are correlated (and that path modeling shows potentially how enzymal activity provides this linkage), and that leaf growth in these systems is nitrogen limited. I did enjoy reading this paper and this work appears to have potential to make a novel and valuable contribution to the literature. I also commend the authors for providing several clear and testable hypotheses in the introduction (lines 97 – 103) and delivering on the promise of addressing how the data supports or refute these hypotheses throughout the results and discussion.

General comments: I recommend this paper for publication following some revisions. Generally, there were instances where claims were made to references to literature in which the text provided did not provide evidence to support the statement. Further, some methodological details remain to be clarified and expanded upon (see below specific comments regarding clarifying regarding whether the C:N:P ratios supplied throughout are molar ratios or mass ratios, or the concerns regarding multiple comparisons).  I also recommend additional & thorough copy-editing throughout due to grammatical mistakes in the manuscript, which make it difficult to understand at times.

Specific Comments:

Line 51: What does ‘gradually increasing’ here mean, in this context? Is this referring to the gradually increasing number of papers published about stoichiometry? The rate of papers published on every topic is increasing, is the rate of papers published about stoichiometry increasing faster than the number of papers published in general?

Line 175-177: How were assumptions of homoscedasticity, linearity, and normality tested before Pearson correlations were performed? There are several hundred correlations being assessed here (Figure 7) - was there any multiple comparison correction (Bonferroni or otherwise) applied? If not, I would recommend considering this approach.

Line 185: This homeostatic index approach is interesting – has this been applied in the context of tree (or specifically, leaf:litter:soil) stoichiometry previously? The approach from the Persson et al. (2010) paper applied to a range of heterotrophs and autotrophs is an interesting framework, but I think some more justification regarding it’s application in the leaf:litter:soil context would be valuable to support it’s inclusion, given the eventual conclusions regarding strict homeostatis between components of these red pine systems resulting from insignificant regressions.

Line 290: I don’t believe it is stated in-text at any point explicitly that these stoichiometric values being reported are molar ratios rather than mass ratios – I presume they are as mol:mol because of the conclusions around nutrient limitations being inferred compared to particular threshold values (e.g., N-limitation at the value of leaf N:P of 6.9; line 407), and because they are being directly compared here to values from the McGroddy et al. 2004 work, which, if I am reading correctly, reports in molar ratios throughout? In any case, it should be made explicit at some point in the paper. The raw mass data, which are currently not directly provided (line 430) would be useful for these applications and for maximum reproducibility of calculation of molar ratios from C/N/P mass contents.  

Line 341: From what I can assess, the two references for this sentence (‘as tree growth progresses, the relative content of C increased dramatically…’) do not substantiate this claim – reference [53] (McJannet et al., 1995) shows a decrease in relative N and P by dry mass with increasing biomass for a set of 41 wetland plants (no tree species, as far as I can tell) grown in pots, and reference [52] (Yang et al., 2011) uses a meta-analysis of primarily growth chamber and open-top chamber manipulation studies to assess changes to C:N stoichiometry associated with elevated CO2 concentration and N addition across many ecosystem and plant types (wetland, grassland, cropland, tundra, and forest). There are perhaps other chronosequence studies (with more comparable/relevant environments & plant types, specifically, trees, or more specifically, Pinus spp. ) which could be referenced to be able to make claims regarding leaf, litter and soil C:N, C:P and N:P ratios varying with age.

Line 430: The raw data are not made available in this article.

Table 1: Was stand density measured, and if so, could it be included here? As well, for clarity, was each of the 23 sites inserted into exactly 1 of the 4 age classes here? If so, could the number of sites which fell into each of the four age classes be included in this table also?

Figure 3: Including error bars to show variability within each age class would be helpful.

Figures 4-6: Some additional details regarding boxplot structure would be helpful – are the black circles shown overlain on the boxes the mean values?

Figure 7: See above comment regarding multiple comparisons.

Author Response

Figure 7 correlation analysis

Reviewer 2 Report

Dear Authors,

Your research work is interesting because it shows the stability of C:N:P stoichiometry in the plant–soil continuum along age classes in natural Pinus tabuliformis forests of the eastern Loess Plateau, China. This paper is prepared in the usual way for scientific work. Manuscript is prepared carefully. However, I have a few comments. Some are debatable:

Line 18. Instead of Pinus tabuliformis (P. tabuliformis), there should be Pinus tabuliformis Carr. (P. tabuliformis).

1.      Lines 37-38. You write, “Further studies are needed to capture the critical factors that regulate leaf stoichiometry in soil system”. Why do you think so when the system is relatively stable (line 36)? What threatens it?

2.      Line 106. Figure 1. Provide the source of the figure or write "own study".

3.      Figure 7. “Correlation analysis” - reduce the font.

In my opinion, the results, discussion and conclusion chapters are well written.

The language appears to be correct, but I don't feel qualified to judge about the English language and style.

I recommend for publication in Forests after the indicated corrections.

Good luck!

Sincerely yours

Reviewer

Round 2

Reviewer 1 Report

I have reviewed the revised manuscript resubmitted by the authors. The authors have done a thorough and thoughtful job carefully responding to the comments of both reviewers. I believe this paper has been improved and is poised to make an important contribution to the literature on this timely and relevant topic. My remaining question with the text is with regards to the author’s response around the application of the homeostatic index (listed as ‘Point 3’ in their response). I believe the authors did an excellent job of justifying the use of the index in their response letter, and believe that some version of this text/explanation should be included within the revised manuscript, perhaps in the methods section, to help guide the reader to the justification of this homeostatic index approach in the forest-soil context. Once this remaining minor issues is fixed, I would recommend this manuscript for publication in Forests.

Author Response

Please see the acttachment
